# Discovery and characterization of a new type of domain wall in a row-wise antiferromagnet

Jonas Spethmann[1], Martin Grünebohm[1], Roland Wiesendanger [1], Kirsten von Bergmann [1✉] & André Kubetzka [1✉]

Antiferromagnets have recently moved into the focus of application-related research, with the perspective to use them in future spintronics devices. At the same time the experimental determination of the detailed spin texture remains challenging. Here we use spin-polarized scanning tunneling microscopy to investigate the spin structure of antiferromagnetic domain walls. Comparison with spin dynamics simulations allows the identification of a new type of domain wall, which is a superposition state of the adjacent domains. We determine the relevant magnetic interactions and derive analytical formulas. Our experiments show a pathway to control the number of domain walls by boundary effects, and demonstrate the possibility to change the position of domain walls by interaction with movable adsorbed atoms. The knowledge about the exact spin structure of the domain walls is crucial for an understanding and theoretical modelling of their properties regarding, for instance, dynamics, response in transport experiments, and manipulation.

[1] Department of Physics, University of Hamburg, Hamburg, Germany. ✉email: kbergman@physnet.uni-hamburg.de; kubetzka@physnet.uni-hamburg.de

Antiferromagnets (AFMs) as first conceived by Louis Néel consist of two identical, interpenetrating ferromagnetic sublattices, magnetized in opposite directions. In his work, he states that AFMs are extremely interesting from the theoretical standpoint but do not appear to have any practical applications[1]. Nowadays, a large variety of magnetic systems with vanishing net magnetization falls into this category, including synthetic AFMs[2,3], collinear[4,5], non-collinear[6–8], and non-coplanar systems[9,10], and AFMs are envisioned to play a prominent role in future spintronic devices[11–13].

AFMs are versatile materials, which display different and generally faster dynamics compared to ferromagnets[14]. They show distinct transport properties which can depend on their topology[15]. These findings have recently inspired investigations that aim at replacing ferromagnetic materials for future applications with AFMs. In particular, the exploitation of movable solitons, like domain walls (DWs) and skyrmions, has been in the focus of research. A number of experimental techniques have been developed and refined to image AFM domains and DW positions, and current-induced DW motion in AFMs has been demonstrated[16]. However, a determination of DW widths and a characterization of the spin configuration within a DW[17] remains a challenging task[18]. Theoretical investigations of AFM DWs have focused on dynamical properties[19], such as Lorentz contraction, suppression of Walker breakdown[20,21], or interactions with spin waves[22,23]. The considered DWs are typically described by a coherent rotation of all magnetic sublattices, i.e., the profiles of these DWs are identical to DWs in ferromagnetic systems[24,25]. However, it has been pointed out that the large variety of AFM states should allow for a wider range of AFM DW configurations compared to ferromagnets[12]. It is expected, that the details of the spin texture within an AFM DW play a crucial role for its properties regarding, for instance, current-induced motion or emerging Hall effects. Therefore it is important both for an understanding as well as a tailoring of AFM DW properties to unveil their spin configuration.

## Results

**The row-wise antiferromagnetic state.** To investigate the details of antiferromagnetic DWs we choose the row-wise anti-ferromagnetic (RW-AFM) state on a hexagonal atomic lattice as model-type system. It is characterized by parallel magnetic moments along a close-packed atomic row (AFM row), with antiparallel alignment between adjacent rows, see sketch in

Fig. 1a. The RW-AFM state is the ground state in the pseudo-morphic fcc-stacked Mn layer on Re(0001), as we have demonstrated previously[10] by a combined spin-polarized scanning tunneling microscopy (SP-STM)[26,27] and density functional theory (DFT) study. It results from an AFM nearest-neighbor Heisenberg exchange coupling, $-J_1(\mathbf{S}_i \cdot \mathbf{S}_j)$ with $J_1 < 0$, and exchange frustration fulfilling $1 > J_2/J_1 > 1/8$[28], where $J_2$ denotes the coupling strength to the next-nearest neighbor. Each magnetic moment has four antiparallel and two parallel neighboring magnetic moments, see sketch, resulting in a contribution of $-2|J_1|$ per atom to the total energy. In a similar fashion, one can derive the contribution of next-nearest neighbor exchange to be $-2|J_2|$. Due to the hexagonal symmetry of the atomic lattice this uniaxial RW-AFM state can occur in three symmetry-equivalent rotational domains, as seen in the overview SP-STM image of Fig. 1c (see Methods). Here the fine lines indicate the antiparallel adjacent AFM rows, see sketches.

A RW-AFM state can be viewed as a uniaxial single spin spiral state (1Q) with a propagation direction perpendicular to the AFM rows. In the Heisenberg model this 1Q state is degenerate with the superposition states of symmetry-equivalent spin spirals, i.e., the 2Q and the 3Q states. Both have the same energy as the RW-AFM state, when only $J_1$ and $J_2$ are considered, as illustrated for the 2Q state in Fig. 1b. In our system, however, apparently a small net contribution from higher-order exchange interactions (HOIs)[29] lowers the energy of the 1Q state with respect to these superposition states[9,10], leading to a RW-AFM ground state. According to DFT calculations[10] the system has an easy-plane crystal anisotropy $-K(S_z)^2$. Because the RW-AFM state is a uniaxial spin texture, it breaks the symmetry of the hexagonal lattice and the total energy depends on the direction of the spins relative to the AFM rows. In the case of our Mn monolayer the spin-orbit interaction related anisotropic symmetric exchange (ASE) energy, as obtained from DFT, favors the spins to be aligned along the AFM rows. This configuration is also preferred by the dipolar interaction, and the schematics in Fig. 1c show the resulting magnetic states.

On a larger scale, see Fig. 1d, DWs can be imaged without atomic-scale resolution by measuring the differential tunnel conductance, which reflects the local density of electronic states. Both the variation of the quantization axis and the change in the nearest-neighbor angle can in general lead to modifications of these electronic states[30,31], see also Supplementary Fig. S1 . Thus the distinct spin configuration within DWs makes them

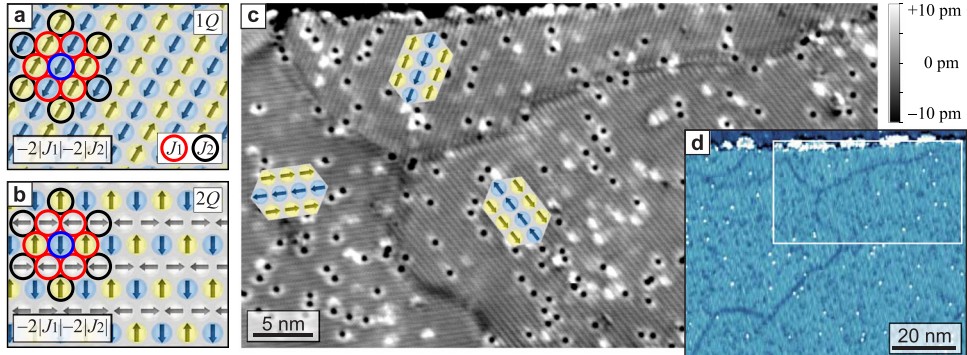

**Fig. 1 The RW-AFM state in the fcc-stacked Mn monolayer on Re(0001), see Methods for sample preparation. a** Sketch of the RW-AFM state, the circles mark nearest- and next-nearest magnetic moments of one exemplary spin. **b** Respective sketch of the 2Q state. **c** Constant-current SP-STM image (height data) exhibiting all three possible rotational domains of the RW-AFM state ($U = -30$ mV, $I = 7$ nA). The domains are separated by DWs, which propagate in various directions and seem to be influenced by the position of atomic defects. **d** Differential tunnel conductance map of a larger field of view ($U = +500$ mV, $I = 3$ nA), the white rectangle indicates the sample area shown in **c**. The AFM spin texture is not resolved, but the DWs are imaged as darker lines due to differences in the electronic states (see Supplementary Fig. S1 for more details).

electronically different compared to the collinear domains, enabling an experimentally convenient detection of DW positions.

**Domain walls in the RW-AFM.** In the uniaxial RW-AFM on a hexagonal atomic lattice different kinds of domain walls can occur. The sample area in Fig. 1c shows several domain walls between rotational domains, i.e. orientational DWs. DWs between two phase-shifted domains of the same rotation are very rare in this system of the fcc Mn monolayer on Re(0001). They connect the translational AFM domains by a coherent 180° spin rotation of the sublattices, as previously found in Fe/W(001)[17]. For our system, such a phase DW is shown in Supplementary Fig. S2, and it runs perpendicular to the AFM rows.

In orientational DWs, as seen in Fig. 1, two different highly symmetric cases are obtained when the DW path is along the bisecting line of the adjacent AFM rows. Then either an angle of 120° occurs at the DW position, or the AFM rows of adjacent domains enclose an angle of 60°. We refer to these highly symmetric orientational DWs as 120° and 60° DWs in the following. Locally the DW orientations vary and apparently they are influenced by native atomic defects. We find that the majority of DWs are closer to 120° DWs, and 60° DWs are less common.

To identify the details of the spin configuration within orientational DWs we measure them with high spatial resolution. Figure 2a shows a closer view of a DW, the bright spots are adsorbed Co atoms. Because the path of the DW is changing at the position of the Co adatoms, it consists of both types of highly symmetric DWs, i.e. a 120° and a 60° DW section. Both show a distinct substructure. Figure 2b shows another 60° DW, imaged under identical conditions and with the same tip. This DW is wider, probably as a result of the position of native defects, some of which are marked by red circles. In this more extended transition region between the rotational domains one can clearly see a hexagonal pattern.

These measurements show that it is challenging to precisely determine the intrinsic DW width and shape from SP-STM measurements: their appearance depends on the DW orientation and seems to be affected by defects, leading to variations in the regime of 1.5–2.5 nm for 60° DWs, see Fig. 2a, b. Likely the tip magnetization direction, which determines the SP-STM contrast, also plays a role for the apparent width of the DWs.

To explore the DWs in the Mn layer from a different perspective we employ magnetic atom manipulation imaging[32,33] in the sample area indicated by the red square in Fig. 2b, including the 60° DW, and the resulting data is shown in Fig. 2c. With this technique, the tip is used to move a magnetic adatom across the surface while scanning, see inset. The adatom typically does not follow the tip continuously, but it jumps from one favorable adsorption site to the next. This leads to triangular plaquettes in the images, which mark the positions of hollow sites in which the adatom resides[34,35], see Methods.

In our case, with a magnetic surface, magnetic adatom, and magnetic tip, the size and shape of the plaquettes is in addition governed by the magnetic interactions, resulting also in a spin-dependent apparent height. This allows to clearly resolve the two rotational RW-AFM domains in Fig. 2c. They appear very differently, because the angle between tip magnetization direction and the quantization axis of the RW-AFM changes, just as in SP-STM images. The spin-dependent height difference for the right domain amounts to about 60 pm, compared to about 7 pm magnetic corrugation amplitude for the SP-STM data in Fig. 2a, b. We also observe a difference in the size of the triangular plaquettes, as indicated by the blue and yellow shapes in Fig. 2d. This means that the lateral positions, at which the Co atom jumps

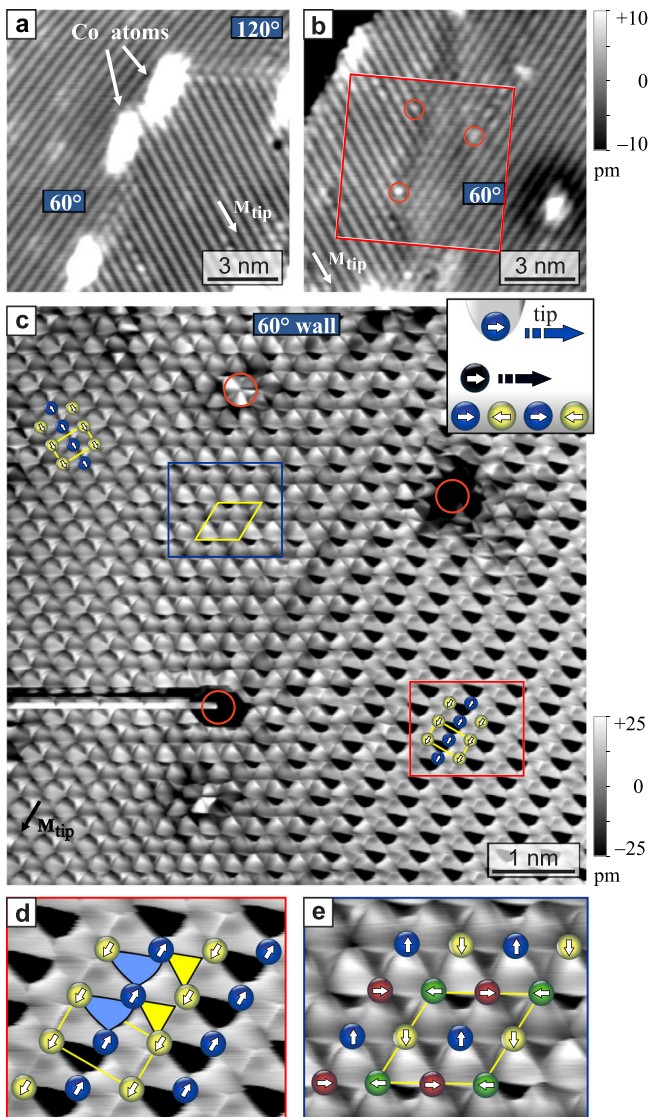

**Fig. 2 Domain walls in the RW-AFM. a** SP-STM image of a DW with Co atoms or clusters with a 60° and a 120° DW section and (**b**) a neighboring sample area about 50 nm away with a 60° DW imaged with the same tip (both: $U = +14.8$ mV, $I = 5$ nA); red circles indicate native defects. **c** Atom manipulation image (height data) of the area indicated by the red frame in (**b**), using a Co atom that moves with the tip, see inset ($U = +3.8$ mV, $I = 60$ nA). Magnetic unit cells are indicated by yellow rectangles and a diamond. **d** Enlarged view of the RW-AFM state indicated by the red frame in (**c**), the schematics of the spin structure is overlaid and the triangular shapes indicate triangles with large magnetic contrast. **e** Enlarged view of the DW region indicated by the blue frame in (**c**), overlaid with a schematic of the 2Q state.

to the next hollow site, alternate. We conclude that the force necessary to move the Co atom is spin-dependent[32].

The transition region, which shows a hexagonal pattern in SP-STM images, has a width of about 3 nm in the manipulation image and can now be clearly identified as a $p(2 \times 2)$ magnetic superstructure; the yellow diamond marks the magnetic unit cell which contains four atoms. This change of symmetry immediately demonstrates that the spin texture within the DW is a different magnetic state, distinct from the RW-AFM domains. Because the two adjacent domains seem to continuously transform into one another, we propose that the DW is equivalent to a superposition of the two adjacent 1Q states, i.e., in the center the

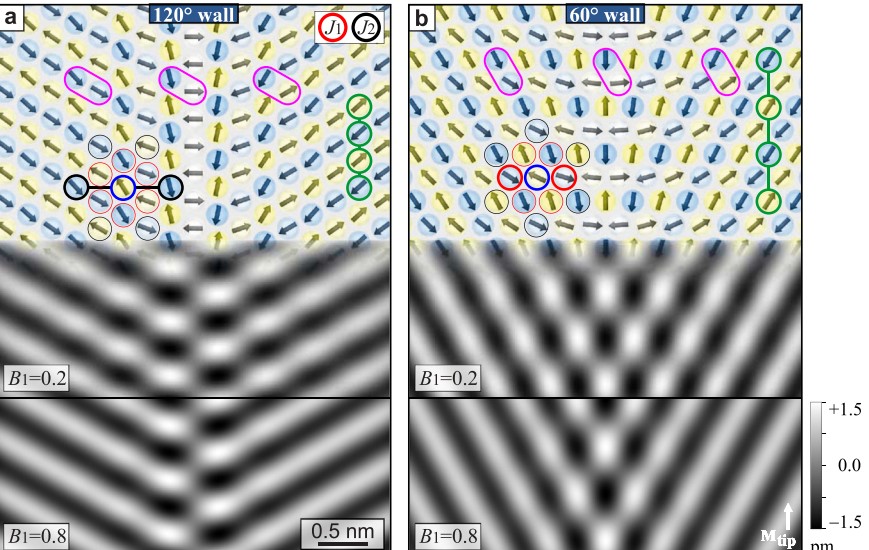

**Fig. 3 Simulations of DWs. a, b** Simulation results for a 120° and a 60° DW using a set of simplified DFT parameters (in meV/atom): $J_1 = -25$, $J_2 = -5$, $J_{ASE} = +0.025$, $K = -1$, and $B_1$ as indicated. At the top the spin structure is depicted and the gray-scale images show SP-STM simulations for the indicated tip magnetization direction $\mathbf{M}_{tip}$ for two different values of $B_1$. The colored circles are explained in the text. For the SP-STM simulations a spin-polarization of the tunnel current of $P = 0.4$, a tip-sample distance of $z = 0.8$ nm, and a work function of $\Theta = 4.8$ eV were used.

DW has the spin structure of a 2Q state, see Fig. 1b and the overlaid spin structure in Fig. 2e. This state also has close-packed atomic rows with antiparallel alignment of neighboring magnetic moments, but adjacent rows have a 90° rotated magnetic quantization axis.

**Simulations of DWs.** To obtain a better understanding of the two highly symmetric orientational DWs we perform atomistic spin dynamics simulations[36] using the following Hamiltonian (see also Supplementary Fig. S3 for more details on the simulations):

$$H = -J_1(\mathbf{S}_i \cdot \mathbf{S}_j) - J_2(\mathbf{S}_i \cdot \mathbf{S}_j) - K(S_z)^2 \\ - J_{ASE}(\mathbf{S}_i \cdot \mathbf{d}_{ij})(\mathbf{S}_j \cdot \mathbf{d}_{ij}) - B_1(\mathbf{S}_i \cdot \mathbf{S}_j)^2 \tag{1}$$

where $\mathbf{d}_{ij}$ is the unit vector pointing from $\mathbf{S}_i$ to $\mathbf{S}_j$. For $J_1$, $J_2$, $K$, and $J_{ASE}$ we use values very close to the ones obtained by DFT for this system[10]. The last term is the biquadratic exchange, or two-site fourth-order HOI, and $B_1$ is chosen to be positive in order to realize the experimentally observed 1Q state. The simulated spin configurations in Fig. 3 for $B_1 = +0.2$ meV/atom show that adjacent horizontal rows rotate in opposite directions across the domain wall, see pink ellipses for selected atom pairs in adjacent rows. This leads to spin configurations with 90° between neighboring magnetic moments in the center of the DWs, which is a characteristic of the 2Q state, Fig. 1b. Thus the simulations confirm the spin structure sketched in Fig. 2e.

To visualize the expected SP-STM results we calculate SP-STM images[37] and show them in Fig. 3 for two different values of $B_1$. The appearance of the DWs is in agreement with the hexagonal pattern in the experimental SP-STM images (Fig. 2). We conclude that the orientational DWs between two rotational domains of the RW-AFM are formed by a superposition of the adjacent 1Q spin textures, resulting locally in a 2Q structure. This new type of superposition DW is qualitatively different from phase domain walls, where the spins of the two sublattices rotate coherently. Considering that for small values of the HOIs the 1Q state and the 2Q state are nearly degenerate, and that the exchange interaction cannot favor one over the other, it seems natural that the DWs form a superposition state. Bearing this in mind one can

easily rationalize the finding that the wall width depends on the strength of the HOI parameter $B_1$, see Fig. 3.

To understand the origin and the properties of this new type of DW we closely inspect the role of the different magnetic interaction parameters. Looking at the atom pairs indicated by the pink ellipses in Fig. 3, one can see how the contribution from the biquadratic term $B_1$ changes across the DW: in the domains all adjacent spins are in a favorable collinear configuration, whereas in the center of the DW four adjacent spins are close to orthogonal with maximal energy cost. In our simulations[36], the three different fourth-order HOI terms[29], i.e. two-site ($B_1$)[38], three-site[5], and four-site[39] four-spin interactions, qualitatively have the same impact on the spin texture within the DWs. We find that in a 1Q–2Q transformation, as present here when going from the 1Q domains to the 2Q wall, all terms scale with $\cos^2(2\alpha)$, see ref. [40] and Supplementary Fig. S4. Consequently, to describe the DWs we can condense the impact of the three HOIs into one effective HOI term, justifying the reduced Hamiltonian of Eq. (1). In the simulations the $\cos^2(2\alpha)$ scaling of the HOIs leads to DW profiles closely following tanh functions, see Supplementary Figs. S5–7. This is reminiscent of ferromagnetic DWs, in which, however, it is the crystal anisotropy that leads to the tanh shape of the DW profiles.

Now we inspect the role of the nearest and next-nearest neighbor exchange interaction. The magnetic moments in rows parallel to the wall (see green circles) are always strictly antiferromagnetic, both in the domain and in the DW. This immediately shows that both DWs have a vanishing net magnetic moment. In Fig. 3 the nearest and next-nearest neighbors of exemplary atoms (blue circles) are indicated by red and black circles, respectively. First we inspect the 120° DW in Fig. 3a: the two neighboring spins above and below a given spin are both strictly antiparallel to it, leading to the identical energy of $-2|J_1|$ for a spin in a domain or in a DW; the two nearest-neighbor spins on the left side of a given atom are antiparallel to each other, leading to opposite energy contributions of exchange interaction with the considered spin, and thus their contributions exactly cancel; the same is true for the two nearest-neighbor spins on the other side. Because its energy contribution does not change as we move across the wall, $J_1$ does not have an impact on the DW

energy or its width. Using the same arguments we find, however, that the situation is different for $J_2$: the energy per atom within the DW is higher than within a domain, because in the DW the angle is less than 180° for the two next-nearest neighbor spins that are indicated by the thick black circles. The situation of $J_1$ and $J_2$ is reversed in the 60° DW, cf. Fig. 3b, where the nearest-neighbor moments indicated by the thick red circles have angles of less than 180°, resulting in a dependence of the DW energy and width on $J_1$.

To summarize: the energy and width of the DWs depend very differently on the exchange interactions for the two highly symmetric DW types. 120° DWs only depend on $J_2$ and 60° DWs only depend on $J_1$. Based on these findings we propose an analytical description, see Table 1, and confirm its validity with numerical simulations (Supplementary Figs. S5–S7). In fcc Mn/Re (0001) – with $J_1/J_2 \approx 4.5$ – the 120° DWs have a lower energy and are narrower compared to the 60° DWs, both by a factor of $\sqrt{4.5/3} \approx 1.2$. This is in agreement with our experimental observation that 120° DWs are predominant and that 60° DWs are broader. Only for $J_1/J_2 < 3$ the 60° DWs would be preferred.

**The magnetic state and the rim**. The number of DWs in our system depends on the size and shape of the Mn film. This originates from an impact of the rim on the rotational domain next to it, which becomes apparent in the irregularly shaped islands shown in Fig. 4. In both Mn areas, all three rotational domains are present. In each image, one rotational domain does not clearly show the typical stripes. We conclude that within those rotational domains all magnetic moments are perpendicular to the respective tip magnetization direction, which is indicated. The black

features are holes in the Mn monolayer. The SP-STM image in Fig. 4a dominantly shows one rotational domain, but it becomes evident that the rim of the hole induces the formation of two additional domains. Also the rotation of the RW-AFM domains in Fig. 4b is governed by the rim of the Mn monolayer. Our experimental data reveal a general trend that the AFM rows avoid to be parallel to a Mn edge. The small area in Fig. 4a, which is marked by the red frame, is an exception, and below this region we find a different rotational domain that again avoids AFM rows parallel to the rim of the vacancy island.

In a simplified model with structurally perfect edges along close-packed rows two different configurations are possible: one of them has the AFM rows at an angle of ±120° to the edge, see Fig. 4c, the other one has AFM rows parallel to the edge, see Fig. 4d. To obtain the energy difference between them we consider the interaction parameters $J_1$ and $J_2$ and assume that they do not change at the edge. In the upper configuration (Fig. 4c) each rim atom (indicated exemplarily by the green circle) has four nearest neighbors with one parallel magnetic moment and three antiparallel magnetic moments. Thus its contribution to the total energy is $-2|J_1|$. The magnetic interactions to the three next-nearest neighbor spins (one parallel and two antiparallel) sum up to $-|J_2|$. For an atom in the second row (purple circle) the analogous reasoning can be done for the six nearest and five next-nearest neighbors. The contribution of atoms in the third row to the total energy are independent of the rim configuration; because we are interested in the energy difference between the two rim configurations we do not need to take them into account. For the configuration sketched in Fig. 4d, with the AFM rows parallel to the rim, the contributions from atoms in the first and second atom row are indicated on the left and right side, respectively. By comparing the individual exchange interaction contributions we can derive the energy difference between the two configurations and obtain $-2|J_1| + 2|J_2|$ per atomic distance. Because in a RW-AFM on a hexagonal lattice the absolute value of $J_1$ is always larger than $J_2$ the rotation of a domain at the rim as shown in Fig. 4c is favored over the one displayed in Fig. 4d, in agreement with the experimental observations.

**Impact of adatoms on DW position**. In the atom manipulation measurements (Fig. 2c) we have seen that the presence of the DW affects the movement of a Co atom. In Fig. 5 we demonstrate the inverse case: the positions of DWs can be changed by manip-

**Table 1 RW-AFM superposition DW widths and energies in small angle approximation and for $B_1 \ll J_1$ (see Supplementary Material).**

| DW type | Model | Width | Energy |
|---|---|---|---|
| 180° FM | A, K (continuum) | $2\sqrt{A/K}$ | $4\sqrt{AK}$ |
| 180° FM | $J_1 > 0, K > 0$ | $2a\sqrt{\frac{3}{2}J_1/K}$ | $\frac{4}{a}\sqrt{2J_1K}$ |
| 120° AFM | $J_1, J_2 < 0, B_1 > 0$ | $\frac{a}{2}\sqrt{3|J_2|/B_1}$ | $\frac{8}{a}\sqrt{|J_2|B_1}$ |
| 60° AFM | $J_1, J_2 < 0, B_1 > 0$ | $\frac{a}{2}\sqrt{|J_1|/B_1}$ | $\frac{8}{a}\sqrt{\frac{1}{3}|J_1|B_1}$ |

We use a minimal model with only $J_1$, $J_2$, and $B_1$ (all in meV/atom) on a hexagonal layer with lattice constant $a$. 180° FM (ferromagnetic) DWs are shown for comparison, with easy-axis anisotropy, $K > 0$, in meV/atom and exchange stiffness A and anisotropy K in SI units.

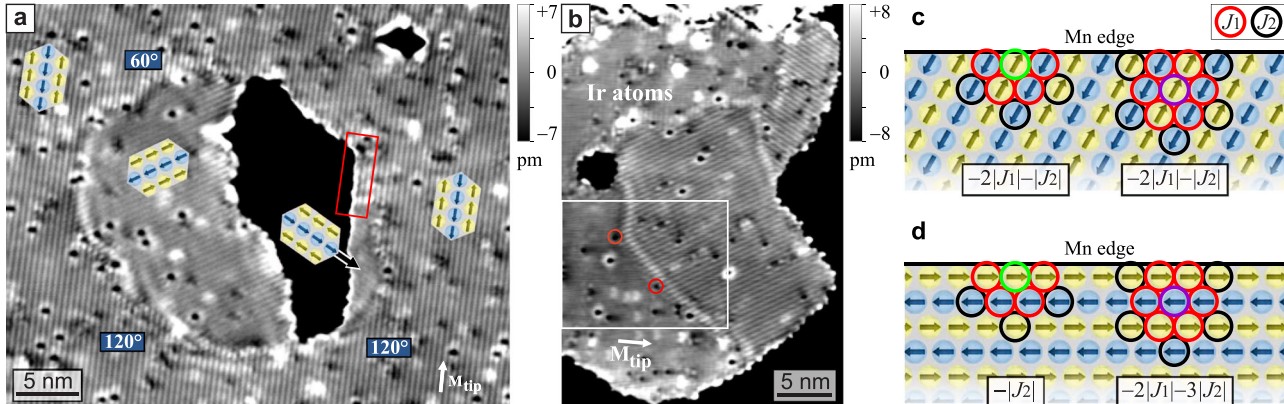

**Fig. 4 Coupling of the RW-AFM to the film boundary. a, b** Constant-current SP-STM images with all three rotational domains. In each measurement one rotational domain shows vanishing magnetic contrast, the derived tip magnetization directions **M$_{tip}$** are indicated. The black areas are holes in the Mn layer (**a**: $U = +10$ mV, $I = 2$ nA; **b**: $U = +50$ mV, $I = 7$ nA). **c, d** The two possible highly symmetric RW-AFM configurations at a close-packed row rim: the AFM rows either enclose ±120° or 0° with the film edge.

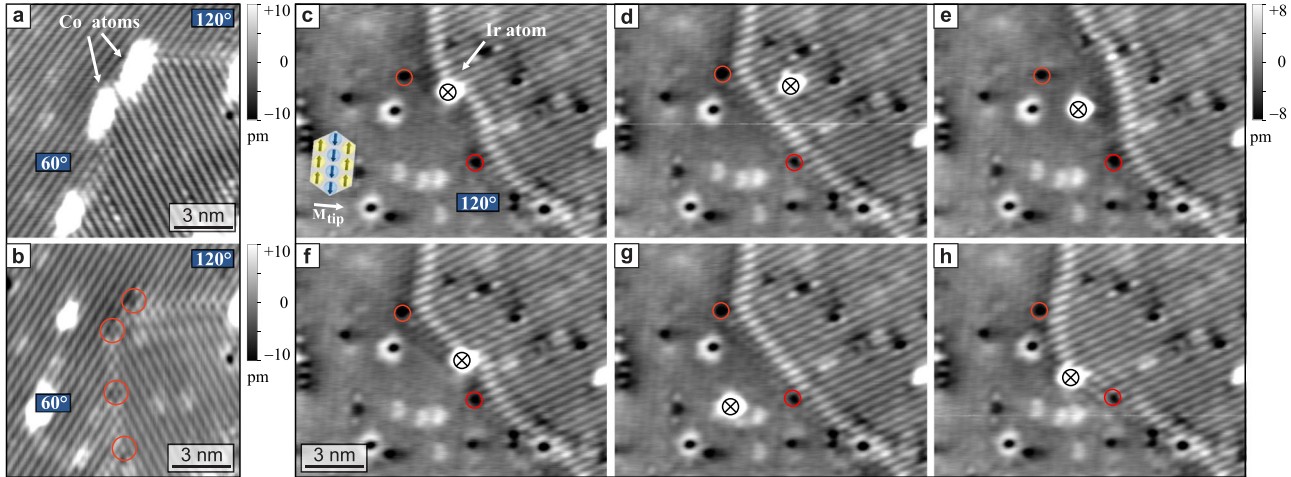

**Fig. 5 Domain wall control by atom manipulation. a** SP-STM image of a DW decorated with Co atoms or clusters, same as shown in Fig. 2a ($U = 14.8$ mV, $I = 5$ nA). **b** When the Co atoms are removed via atom manipulation, the DW relaxes to a slightly different position, pinned to native defects (red circles); the tip has changed in the manipulation process ($U = +19.8$ mV, $I = 5$ nA, atom manipulation: $U = +4$ mV, $I = 40$ nA). **c–h** Consecutive images of the same sample area with an Ir adatom, which is indicated by the cross. The DW position changes when the Ir atom is moved (images: $U = +10$ mV, $I = 2$ nA, manipulation parameters: $U = +3$ mV, $I = 60$ nA).

ulating individual atoms. The SP-STM image of Fig. 2a is again displayed in Fig. 5a. When the Co atoms or clusters, that decorate the DW, are removed via atom manipulation, see Fig. 5b, the DW relaxes to a slightly different position. The 60° DW section has moved to the right, and the 120° DW section deviates significantly from the high symmetry direction and appears more complex. The tip has also changed in the manipulation process and the relative magnetic corrugation amplitudes of the two RW-AFM domains differ for the two measurements. This pair of images demonstrates that DWs interact with and can be pinned by both native defects, as indicated by red circles in Fig. 5b, and deposited adatoms.

To explore this further, we use Ir as an adatom, which is non-magnetic as bulk material. In general, pinning does not necessarily require a magnetic moment but can be achieved by a local change of one of the magnetic interactions[41]. The sample area shown in Fig. 5c–h is marked in Fig. 4b, and for the left domain the magnetic contrast almost vanishes for this particular tip magnetization direction, see schematics in Fig. 5c. In this series of consecutive images we observe that the position of an Ir adsorbate (black cross) influences the path of the DW. In panels d and g the DW is in its unperturbed position, which is likely governed by the position of native defects, two of which are marked by red circles. However, in the other images we find that different positions of the Ir adatom lead to different distortions of the DW, compare Fig. 5c, e, f, h.

## Discussion

In the model-type system of a hexagonal Mn monolayer on Re (0001), which has the RW-AFM as magnetic ground state, we have identified and characterized a novel type of domain wall in antiferromagnets, which consists of a superposition state of the adjacent domains. This newly established type of DW is qualitatively different from phase domain walls, where the spins of the two sublattices rotate coherently.

The width and energy of these superposition DWs is governed by a balance of magnetic exchange interactions and higher-order interactions. Depending on the relative orientation of the adjacent rotational domains two highly symmetric DWs are possible, and analytical equations are derived. Interestingly, each of the two symmetric DWs depends on only one of the exchange interactions—nearest- or next-nearest neighbor exchange—whereas the

other one does not contribute to its energy or width. In the limit of vanishing HOIs, the RW-AFM and the 2Q state become degenerate.

The non-collinear spin texture of the wall gives rise to a modification of the electronic states with respect to the collinear magnetic state, enabling a detection of DWs in differential tunnel conductance images at larger scales. Further implications of the non-collinearity of the wall include distinct interactions with lateral currents, such as effects related to the chirality, with possibly large DW signals in transport measurements[15].

We have discovered that the RW-AFM state couples to the boundary of the magnetic material: rotational domains with AFM rows canted relative to the edge are preferred over those with AFM rows parallel. We have explained this in a simple model considering the contributions of nearest- and next-nearest neighbor magnetic interactions. Because the selection of a specific rotational domain by the rim often induces the incorporation of DWs, the cost of a DW must be similar or smaller than the energy difference between the two edge configurations of Fig. 4. This preferential coupling of the spin texture to the rim is similar to closure domain formation in ferromagnets at boundaries to reduce the stray field. It can be exploited to generate a significant number of DWs in an antiferromagnet.

The path of the DWs is influenced not only by native defects but also by magnetic or non-magnetic adsorbates. By employing atom manipulation this offers the opportunity to prepare and tailor specific AFM spin configurations that provide a playground for further studies. In this way, deeper insight into the magnetic properties of AFM systems might be possible. Furthermore, DW control by atom positioning could, for instance, also facilitate the investigation of the interplay of complex spin textures with the superconducting Re substrate below $T = 1.7$ K[42,43].

Finally, we have demonstrated that indeed the large variety of AFM states –or, to be more general, states with locally compensated magnetic moments– can give rise to unexpected new types of domain walls with intriguing spin textures. The question arises whether the superposition DWs found here are expected to arise also in other systems. Of course there are antiferromagnets, in which a superposition state is not possible. This is for instance the case for spin textures that have the same symmetry as the lattice, e.g. the checkerboard antiferromagnet on a square lattice[17]. In contrast, a uniaxial RW-AFM state on a square lattice can form a

superposition state[44] and is likely to exhibit this new type of DW, see Supplementary Fig. S8. The layer-wise AFMs in bulk materials can be viewed as counterparts of the two-dimensional RW-AFMs, and when domains with an angle between the ferromagnetic planes connect, the resulting DWs are likely candidates for superposition DWs as well. Further investigations will show, how common such superposition DWs are, which other kinds of complex DWs can emerge, and how a specific DW spin texture influences important application-related properties like energy, width, or transport.

## Methods

**Sample preparation.** The experiments were performed in a multi-chamber ultra-high vacuum system with different chambers for substrate cleaning, metal deposition, and scanning tunneling microscopy (STM) measurements. The Re (0001) single crystal surface was cleaned by cycles of annealing in an oxygen atmosphere of $10^{-7} - 10^{-8}$ mbar at temperatures of up to 1400 K; before metal deposition a final flash to $T = 1800$ K was performed. The Mn was evaporated from a pyrolytic boron nitride (PBN) Knudsen cell of volume 1 cm$^3$, held at $T = 620°C$, resulting in a flux of ~0.1 atomic layers per minute; during Mn deposition the Re single crystal was still at elevated temperature (≈400 K) from the final flash[10]. For manipulation experiments, single Co or Ir atoms were deposited onto the cold sample surface.

**STM measurements.** We use a home-built STM at $T = 4.2$ K, equipped with a Cr bulk tip, which—depending on its in-situ treatment—can assume an arbitrary magnetization direction with varying degrees of spin-polarization. The spin-polarized tunnel current then depends on the projection of tip and sample magnetization, leading to the stripe contrast for the RW-AFM, where the magnetic moments within one stripe are parallel to each other. Because the tip magnetization direction is fixed within one measurement, the three rotational domains of the RW-AFM show different magnetic corrugation amplitudes. When the tip magnetization direction is perpendicular to the quantization axis of one of the rotational domains, the magnetic contrast vanishes, as in Fig. 4.

**Atom manipulation imaging.** The images obtained in atom manipulation imaging mode are more complex than standard SP-STM, because the manipulated atom introduces additional degrees of freedom which contribute non-linearly to the tunnel current. On non-magnetic hexagonal surfaces the adatom can reside in two different types of hollow sites (fcc or hcp). In atom manipulation images they can be discriminated by the orientation of the corresponding triangular plaquettes[34,35], i.e. all triangles pointing in one direction indicate the position of one particular three-fold hollow site, triangles pointing in the other direction are at the position of the other type of hollow site, see also Fig. 2d. When a magnetic adatom is used the residence time in a particular site can depend on the magnetic environment. A magnetic tip can influence the measurement in two ways: it can impose a favored magnetization direction on the adatom by exchange coupling, and it leads to a spin-polarized tunnel current. Therefore, the magnetic adatom can be used as a local sensor and amplifier of the (spin-polarized) tunnel signal.

## Data availability

The data that support the findings of this study are available from the authors upon reasonable request.

## Code availability

The spin dynamics simulations of the manuscript were done with Monte Crystal 3.2.0, which can be found on github[36]. The SP-STM simulations were performed in the spirit of this work[37]. Details are available from the authors upon reasonable request.

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

## Acknowledgements

A.K. and K.v.B. acknowledge financial support from the Deutsche Forschungsgemeinschaft (DFG, German Research Foundation) Grants Nos. 408119516 and 418425860. R.W. acknowledges financial support from the ERC (Adv. Grant ADMIRE). A.K. thanks J. Hagemeister for adding ASE to the simulation code[36] and M. Bazarnik, R. Lo Conte, and B. Wolter for discussions.

## Author contributions

J.S. and A.K. performed the experiments. M.G. and A.K. did the spin dynamics simulations, A.K. derived the analytical DW formulas. A.K. and K.v.B. wrote the manuscript. J.S., M.G., R.W., K.v.B., and A.K. discussed the results and contributed to the manuscript.

## Funding

## Competing interests

The authors declare no competing interests.
