## [Peer Review File · Nature Communications]

Reviewers' Comments:

Reviewer #1:

Remarks to the Author:

The manuscript by Spethmann et al reports a discovery of a new type of orientation domain walls with 2Q structure in two dimensional antiferromagnet Mn/Re(0001) using spin polarized-scanning tunneling microscopy (SP-STM). Via atom manipulation, the authors provided clear evidence of the 2Q structure, and demonstrated the pinning of domain walls by adatoms. The STM results are of high quality and the presentation is clear. The discovery of new orientation domain walls is an important contribution to the antiferromagnetism and magnetism in general. Therefore, I believe this manuscript is commensurate with the high standards of Nature Communications. The authors need to address the following questions in the revised manuscript before publication.

1. In the abstract, the authors suggest possibility of manipulate domain walls with the pinning interaction from adsorbed atoms. However, domain walls were also pinned by native defects (C or O) and the atom manipulation of adsorbed atoms only results in local changes of domain walls. Thus, "manipulate" seems to be too strong a claim. "Perturb" would be more appropriate.
2. The authors used atomic manipulation imaging to enhance the spin polarized contrast in the region of 60 degree domain walls to demonstrate the 2Q spin configuration. I wonder whether the authors tried the same technique on 120 degree domain walls. The simulation seems to suggest 2Q domain wall structure in both 60 and 120 degree domain walls.
3. What is the ordering temperature of the row-wise antiferromagnetism in Mn/Re(0001)? Is it consistent with the energy scale of J_1 and J_2 ?

Reviewer #2:

Remarks to the Author:

The manuscript of Spethmann et al. reports experimental results on domain walls in an antiferromagnetic sample with atomic resolution using spin-polarized STM. Antiferromagnetic materials have gained scientific interest in recent years due to their potential application in spintronic devices. Hereby, one exploits their superior switching properties. Although magnetic domains and domain walls play a crucial role in this research field, the precise configuration of magnetic moments within a domain wall remained a mystery due to the limited spatial resolution of the applied methods. This manuscript reveals for the first time details of the magnetization distribution within an antiferromagnetic domain wall.

The main finding is a novel type of domain wall that can be described as a superposition of two translational symmetric spin spirals with integer wave lengths. The finding is substantiated by a simulation based on nearest and next-nearest neighbor exchange interaction parameters. The model limits the parameter range for these parameters to explain the experimental findings. The model also provides some predictions for other materials, if the exchange interaction parameters are known.

The authors further discuss the magnetization configurations at step edges, resulting in a surprising energetic advantage for domainwalls that are not parallel to the step edge. The manuscript also reports on the possibility to direct domain walls at will using atomic manipulation.

In my opinion, the manuscript presents a remarkable advance in the understanding of magnetic domain walls in antiferromagnets. Therefore, I recommend to accept the manuscript for publication.

A few weaknesses of the manuscript should be ammended before publications:

1. Figure 2: Caption refers to red frames. Since there appear two red frames in the large figures it remains unclear which is which. Please use different colors are marks.

2. Table 1: Neither the main text nor the supplement explains how one obtains the analytical formulas given in Table 1. Please provide a reference or add the information to the main text or to the supplement.

3. Figure 4: In (c) and (d) sketches of the magnetization configuration at a step edge is shown. Even after starring for a long time on the picture I could not understand the derivation of the different energetics following from the different wall direction. Moreover, I do not know what the atoms with the lighter colors at the upper boundary mean. Please explain in mnore detail, how the expressions $-2J_1-J_2$ and so on are derived.

Reviewer #3:

Remarks to the Author:

The manuscript by Spethmann and coworkers reports on a study of domain walls in antiferromagnetically ordered monolayer of Mn on Re(0001) using spin-polarized scanning tunnelling microscopy. They conclude that the domain walls are of a new type where at the domain wall, a superposition of the orders on the two sides exists, and discuss its pinning properties. The paper is overall well written and the data of high quality. It is a bit unclear what the main point is which the authors want to convey: is it the pinning properties, or the discovery of a new type of domain wall? My impression is that the paper would benefit from being focussed on one of the two.

Given that the domain walls are shown to move if atoms are removed from the surface, one wonders whether during the atom manipulation imaging, the manipulation of the atom leads to movement of the domain wall?

Minor issues, which however should be fixed:

- * All images are missing a color bar with height scale (or differential conductance scale, respectively)
- * Corresponding authors should be consistent between supplementary and main manuscript
- * Fig. 2c, d, e – it would be useful to use boxes with different colors for panels d and e and to indicate the respective regions in panel c.
- * It is not surprising that non-magnetic atoms can pin the domain wall (l. 242, 243) – this should be rephrased.

Typos:

Line 269: chirality

Reviewer #4:

Remarks to the Author:

The submitted work by Spethmann et al. reports on spin-polarized scanning tunneling microscopy measurements of antiferromagnetic domain walls, accompanied by spin dynamics simulations. The main finding is the discovery of a new type of domain wall in a monolayer AFM. Antiferromagnetic spintronics is a promising research field and domain walls are an important aspect of it. The quality of the experimental data is outstanding and the data are well presented. The methodology is adequate as is the comparison to spin model simulations. The paper is well written and the experimental part understandable. I have, however, two major concerns.

One regards the interpretation of the experimental results. I wonder whether one could possibly

find alternative spin structures within the domain walls that could explain the measured contrast equally. The interpretation of the measured contrast rests very much on the comparison to the spin model. But the choice of parameters for the spin model appears arbitrary to me and it remains unclear where these come from. The most important reference in that context is Ref [10], from - partially - the same authors. But here, the authors use rather different model parameters, B_1 has even a different sign and J_{ASE} was not calculated in [10].

The other problem concerns the significance in the field of AFM spintronics. A special type of domain wall in a monolayer magnet with very unusual (biquadratic) interactions might have limited significance. For these two reasons I hesitate to recommend this work for publication in Nature Communications.

Further remarks:

- I strongly recommend to add a paragraph on the spin model in the methods section and to define the Hamiltonian of the spin model for a clear comparison to Ref [10].
- In line 138 the mathematical expression should probably contain B_1 not just B .

REVIEWER COMMENTS and point-by-point REPLY

Reviewer #1 (Remarks to the Author):

The manuscript by Spethmann et al reports a discovery of a new type of orientation domain walls with 2Q structure in two dimensional antiferromagnet Mn/Re(0001) using spin polarized-scanning tunneling microscopy (SP-STM). Via atom manipulation, the authors provided clear evidence of the 2Q structure, and demonstrated the pinning of domain walls by adatoms. The STM results are of high quality and the presentation is clear. The discovery of new orientation domain walls is an important contribution to the antiferromagnetism and magnetism in general. Therefore, I believe this manuscript is commensurate with the high standards of Nature Communications. The authors need to address the following questions in the revised manuscript before publication.

We thank the reviewer for evaluating our work and for pointing out its significance to the magnetism community.

1. In the abstract, the authors suggest possibility of manipulate domain walls with the pinning interaction from adsorbed atoms. However, domain walls were also pinned by native defects (C or O) and the atom manipulation of adsorbed atoms only results in local changes of domain walls. Thus, “manipulate” seems to be too strong a claim. “Perturb” would be more appropriate.

We have followed the reviewers' recommendation to remove the word “manipulate” in the abstract and have replaced it with “change”.

2. The authors used atomic manipulation imaging to enhance the spin polarized contrast in the region of 60 degree domain walls to demonstrate the 2Q spin configuration. I wonder whether the authors tried the same technique on 120 degree domain walls. The simulation seems to suggest 2Q domain wall structure in both 60 and 120 degree domain walls.

We have also performed atom manipulation imaging across 120° domain walls, and one image is shown below. Indeed, also in this 120° domain wall, a $p(2 \times 2)$ magnetic unit cell becomes visible, as indicated by the yellow diamonds. However, since the 120° walls are thinner compared to 60° walls and the magnetic contrast is lower in this example, we chose to display only the wider 60° domain wall in Fig. 2c,e.

Figure: Atom manipulation imaging with a Co atom across a 120° domain wall ($U=+5$ mV, $I=50$ nA, Cr tip, $T=4.2$ K). The yellow diamonds indicate the $p(2 \times 2)$ unit cell in the center of the wall.

3. What is the ordering temperature of the row-wise antiferromagnetism in Mn/Re(0001)? Is it consistent with the energy scale of J1 and J2?

The ordering temperature of Mn/Re(0001) has not yet been determined, neither experimentally nor theoretically, and our low temperature STM is not an ideal tool to tackle this question. We have, however, made a small set of spin dynamics simulations using the DFT-derived magnetic parameters, and find that the critical temperature is on the order of 130 K, see the figure below.

Figure: Temperature-dependent total energy E and specific heat C of fcc-Mn/Re(0001) obtained from a spin dynamics simulation using a layer of 256 x 256 spins.

Reviewer #2 (Remarks to the Author):

The manuscript of Spethmann et al. reports experimental results on domain walls in an antiferromagnetic sample with atomic resolution using spin-polarized STM. Antiferromagnetic materials have gained scientific interest in recent years due to their potential application in spintronic devices. Hereby, one exploits their superior switching properties. Although magnetic domains and domain walls play a crucial role in this research field, the precise configuration of magnetic moments within a domain wall remained a mystery due to the limited spatial resolution of the applied methods. This manuscript reveals for the first time details of the magnetization distribution within an antiferromagnetic domain wall.

The main finding is a novel type of domain wall that can be described as a superposition of two translational rotational!? symmetric spin spirals with integer wave lengths. The finding is substantiated by a simulation based on nearest and next-nearest neighbor exchange interaction parameters. The model limits the parameter range for these parameters to explain the experimental findings. The model also provides some predictions for other materials, if the exchange interaction parameters are known.

The authors further discuss the magnetization configurations at step edges, resulting in a surprising energetic advantage for domain walls that are not parallel to the step edge. The manuscript also

reports on the possibility to direct domain walls at will using atomic manipulation. In my opinion, the manuscript presents a remarkable advance in the understanding of magnetic domain walls in antiferromagnets. Therefore, I recommend to accept the manuscript for publication.

We thank the reviewer for carefully reading our manuscript and for the appreciation of our work.

A few weaknesses of the manuscript should be amended before publications:

1. Figure 2: Caption refers to red frames. Since there appear two red frames in the large figures it remains unclear which is which. Please use different colors as marks.

We have followed the reviewer's advice and now use different colors for the frames in Fig. 2.

2. Table 1: Neither the main text nor the supplement explains how one obtains the analytical formulas given in Table 1. Please provide a reference or add the information to the main text or to the supplement.

We thank the reviewer for pointing out that this was not clear enough in the last version. We have rephrased the corresponding paragraphs in the revised manuscript and now explain the role of the different interactions for the domain wall width and energy in a clearer fashion (lines 165 - 205 in the revised manuscript). Accordingly, we have changed the order in the corresponding sentence in the abstract.

3. Figure 4: In (c) and (d) sketches of the magnetization configuration at a step edge is shown. Even after staring for a long time on the picture I could not understand the derivation of the different energetics following from the different wall direction. Moreover, I do not know what the atoms with the lighter colors at the upper boundary mean. Please explain in more detail, how the expressions $-2J_1 - J_2$ and so on are derived.

Based on the reviewer's comment we have modified Fig. 4c,d to make the position of the rim and the relative rotation of the RW-AFM state clearer. We have also rephrased the corresponding text (lines 224 - 237 in the revised manuscript). We realized that the energy difference we stated in the previous version was a factor 2 too small. We have corrected this, however, the conclusions remain unchanged.

Reviewer #3 (Remarks to the Author):

The manuscript by Spethmann and coworkers reports on a study of domain walls in antiferromagnetically ordered monolayer of Mn on Re(0001) using spin-polarized scanning tunnelling microscopy. They conclude that the domain walls are of a new type where at the domain wall, a superposition of the orders on the two sides exists, and discuss its pinning properties. The paper is overall well written and the data of high quality. It is a bit unclear what the main point is which the authors want to convey: is it the pinning properties, or the discovery of a new type of domain wall? My impression is that the paper would benefit from being focused on one of the two.

We thank the reviewer for assessing our work. Indeed, both the discovery of a new type of antiferromagnetic domain wall as well as the demonstration of pinning and movement of the walls are addressed in this manuscript. While the new wall type is the novel physics aspect, which is highly relevant for applications such as domain wall movement, also the pinning properties of such domain walls and in particular control thereof play a pivotal role for technical exploitation of antiferromagnets. Therefore, we have included both topics in one manuscript, at the same time avoiding the submission of smallest publishable units.

Given that the domain walls are shown to move if atoms are removed from the surface, one wonders whether during the atom manipulation imaging, the manipulation of the atom leads to movement of the domain wall?

A domain wall movement during atom manipulation imaging is indeed possible and sometimes observed. To characterize the details of the domain wall (Fig. 2c) we have selected data where the domain wall does not change position during scanning. By comparing trace and retrace scan we can confirm that this domain wall does not move, probably as a result of the surrounding defects.

Minor issues, which however should be fixed:

* All images are missing a color bar with height scale (or differential conductance scale, respectively)

We have added the respective color bars in the revised version.

* Corresponding authors should be consistent between supplementary and main manuscript

We disagree, Jonas Spethmann has been in charge of coordinating the Supplemental Material. We would like to acknowledge this contribution and at the same time provide his e-mail address for correspondence in case questions arise.

* Fig. 2c, d, e – it would be useful to use boxes with different colors for panels d and e and to indicate the respective regions in panel c.

We have followed this suggestion and now use different colors.

* It is not surprising that non-magnetic atoms can pin the domain wall (l. 242, 243) – this should be rephrased.

We agree with the reviewer that this statement, together with its withdrawal in the following sentence, should be omitted. We have removed this in the revised version and added the following sentence in lines 253-254: "*In general pinning does not necessarily require a magnetic moment but can be achieved by a local change of one of the magnetic interactions [42]*".

Typos:

Line 269: chirality

We thank the reviewer for pointing this out.

Reviewer #4 (Remarks to the Author):

The submitted work by Spethmann et al. reports on spin-polarized scanning tunneling microscopy measurements of antiferromagnetic domain walls, accompanied by spin dynamics simulations. The main finding is the discovery of a new type of domain wall in a monolayer AFM. Antiferromagnetic spintronics is a promising research field and domain walls are an important aspect of it. The quality of the experimental data is outstanding and the data are well presented. The methodology is adequate as is the comparison to spin model simulations. The paper is well written and the experimental part understandable. I have, however, two major concerns.

One regards the interpretation of the experimental results. I wonder whether one could possibly find alternative spin structures within the domain walls that could explain the measured contrast equally.

This is an important question, and we have repeatedly thought about it. What we see in the experiments (Fig. 2) is that the magnetic unit cell changes from rectangular with two atoms in the magnetic unit cell, to diamond-shaped (or triangular or hexagonal) with four atoms in the magnetic unit cell. We do not have the full vectorial information about the individual spin directions of each of the four atoms, but the change of symmetry from rectangular in the domains to $p(2 \times 2)$ indicates a distinct non-collinear spin texture within the wall (beyond the non-collinearity necessary to go from one collinear domain to the adjacent one). Two non-collinear states, which are in agreement with the observed magnetic unit cell, come to mind: the 2Q and the 3Q state, and also all distortions between them, as they can be transformed into each other continuously. Based on the experiments alone, we cannot single out one of these related magnetic states, which is why we write "we propose that the DW is equivalent to a superposition of the two adjacent 1Q states" (line 136-137 of the revised manuscript). The reason for this proposal is the fact that for a given effective fourth order term, the energy of the 2Q state is always in between the energies of the 1Q state and the 3Q state. For a system with a 1Q ground state and $E(1Q) = 0$ one can derive $E(2Q) = \frac{3}{4} E(3Q)$, making the 2Q state the more probable candidate for our $p(2 \times 2)$ structure. In addition, the easy plane anisotropy also favors the coplanar 2Q state over the 3Q state.

To improve our argumentation in the main text without making it more complicated we have emphasized the symmetry argument (lines 133 - 136 in the revised manuscript).

The interpretation of the measured contrast rests very much on the comparison to the spin model. But the choice of parameters for the spin model appears arbitrary to me and it remains unclear where these come from. The most important reference in that context is Ref [10], from - partially - the same authors. But here, the authors use rather different model parameters, B_1 has even a different sign and J_{ASE} was not calculated in [10].

We thank the reviewer for this comment, and we have taken the opportunity to improve on this part of the manuscript. At the beginning of the simulation section (lines 143 - 148 in the revised manuscript) we have now introduced the Hamiltonian used in the simulations, and explained the choice of parameters ("We use values J_1 , J_2 , K , and J_{ASE} very close to the ones obtained by DFT for this system [Ref Spethmann PRL 2020]. The last term is the biquadratic exchange, or two-site fourth-order HOI, and B_1 is chosen to be positive in order to realize the experimentally observed 1Q state."). In the referenced manuscript indeed the sign of the B_1 parameter was calculated to be

small and negative, and the 3Q state was slightly lower in energy than the RW-AFM state; however, because we observe a RW-AFM ground state we use the positive B_1 as an effective HOI to model our system.

Also J_{ASE} was calculated in that reference: the energy difference between parallel and perpendicular spins is 0.1 meV (main text) and thus $J_{ASE} = 0.025$ meV (supplement); this is the value we use in the simulations of the present work.

The other problem concerns the significance in the field of AFM spintronics. A special type of domain wall in a monolayer magnet with very unusual (biquadratic) interactions might have limited significance. For these two reasons I hesitate to recommend this work for publication in Nature Communications.

To convince also this reviewer of the significance of our work, we have slightly expanded the introduction and the discussion section to emphasize the importance of the spin texture for current-induced DW motion and also to explain our expectation that such DW can occur in a large class of AFM materials.

However, in direct relation to the reviewer's comment we want to highlight two aspects here:

First, our results are important because they demonstrate that DWs in AFMs can be more complex than previously conceived. The example shown here might be one out of many other DW spin configurations in (more) complex AFMs to be discovered in the future.

Second, the size of the HOI can be vanishingly small for this superposition DW to occur. As demonstrated, with the correct sign the size of the HOI just governs the DW width and its energy. We anticipate that whenever an AFM spin texture has lower symmetry than the supporting crystal lattice, rotational domains can arise, and the domain walls between these rotational domains cannot be of standard type, i.e. they cannot be mapped on FM domain walls, and will therefore have properties that are not captured by standard models currently used for AFM domain walls.

Further remarks:

- I strongly recommend to add a paragraph on the spin model in the methods section and to define the Hamiltonian of the spin model for a clear comparison to Ref [10].

As mentioned above we have now introduced the model Hamiltonian in the simulation section and explained the choice of parameters. We thank the reviewer for this advice.

- In line 138 the mathematical expression should probably contain B_1 not just B .

We thank the reviewer for pointing this out.

Reviewers' Comments:

Reviewer #1:

Remarks to the Author:

The authors addressed referees' comments and criticism satisfactorily. The manuscript is ready for publication in Nature Communications.

Reviewer #2:

Remarks to the Author:

In my opinion all questions of the referees have been convincingly answered. I recommend the manuscript for publishing in its present state.

Reviewer #3:

Remarks to the Author:

The authors have carefully considered my comments, and I am happy for the manuscript to be accepted as it is.

In regards to the question whether the atom manipulation moves the domain wall in figure 2c, it would be good if the authors could add the backwards scan, for example as supplementary information, or linecuts across the domain wall from forward and backward trace.

Reviewer #4:

Remarks to the Author:

I thank the authors for their reply and clarifications. The improvements they made in the manuscript are fully sufficient to allay my concerns. I recommend publication of the manuscript.

REVIEWERS' COMMENTS

and authors' answers.

Reviewer #1 (Remarks to the Author):

The authors addressed referees' comments and criticism satisfactorily. The manuscript is ready for publication in Nature Communications.

We would like to thank the reviewer for recommending publication of our manuscript in Nature Communications.

Reviewer #2 (Remarks to the Author):

In my opinion all questions of the referees have been convincingly answered. I recommend the manuscript for publishing in its present state.

We would like to thank the reviewer for recommending publication of our manuscript in Nature Communications.

Reviewer #3 (Remarks to the Author):

The authors have carefully considered my comments, and I am happy for the manuscript to be accepted as it is.

In regards to the question whether the atom manipulation moves the domain wall in figure 2c, it would be good if the authors could add the backwards scan, for example as supplementary information, or linecuts across the domain wall from forward and backward trace.

We would like to thank the reviewer for recommending publication of our manuscript in Nature Communications as it is. We decided to not add the backward scan of the manipulation image to the supplementary material as it contains no additional information.

Reviewer #4 (Remarks to the Author):

I thank the authors for their reply and clarifications. The improvements they made in the manuscript are fully sufficient to allay my concerns. I recommend publications of the manuscript.

We would like to thank the reviewer for recommending publication of our manuscript in Nature Communications.